# The TRPA1 Ion Channel Mediates Oxidative Stress-Related Migraine Pathogenesis

**DOI:** 10.3390/molecules29143385

**Published:** 2024-07-18

**Authors:** Michal Fila, Lukasz Przyslo, Marcin Derwich, Piotr Sobczuk, Elzbieta Pawlowska, Janusz Blasiak

**Affiliations:** 1Department of Developmental Neurology and Epileptology, Polish Mother’s Memorial Hospital Research Institute, 93-338 Lodz, Poland; michal.fila@iczmp.edu.pl (M.F.); lukasz.przyslo@iczmp.edu.pl (L.P.); 2Department of Pediatric Dentistry, Medical University of Lodz, 92-217 Lodz, Poland; marcin.derwich@umed.lodz.pl (M.D.); elzbieta.pawlowska@umed.lodz.pl (E.P.); 3Emergency Medicine and Disaster Medicine Department, Medical University of Lodz, 92-209 Lodz, Poland; piotr.sobczuk@umed.lodz.pl; 4Department of Orthopaedics and Traumatology, Polish Mothers’ Memorial Hospital–Research Institute, Rzgowska 281, 93-338 Lodz, Poland; 5Faculty of Medicine, Collegium Medicum, Mazovian Academy in Plock, 09-402 Plock, Poland

**Keywords:** TRPA1, migraine, oxidative stress, reactive oxygen and nitrogen species, calcitonin gene-related peptide

## Abstract

Although the introduction of drugs targeting calcitonin gene-related peptide (CGRP) revolutionized migraine treatment, still a substantial proportion of migraine patients do not respond satisfactorily to such a treatment, and new therapeutic targets are needed. Therefore, molecular studies on migraine pathogenesis are justified. Oxidative stress is implicated in migraine pathogenesis, as many migraine triggers are related to the production of reactive oxygen and nitrogen species (RONS). Migraine has been proposed as a superior mechanism of the brain to face oxidative stress resulting from energetic imbalance. However, the precise mechanism behind the link between migraine and oxidative stress is not known. Nociceptive primary afferent nerve fiber endings express ion channel receptors that change harmful stimuli into electric pain signals. Transient receptor potential cation channel subfamily A member 1 (TRPA1) is an ion channel that can be activated by oxidative stress products and stimulate the release of CGRP from nerve endings. It is a transmembrane protein with ankyrin repeats and conserved cysteines in its N-terminus embedded in the cytosol. TRPA1 may be a central element of the signaling pathway from oxidative stress and NO production to CGRP release, which may play a critical role in headache induction. In this narrative review, we present information on the role of oxidative stress in migraine pathogenesis and provide arguments that TRPA1 may be “a missing link” between oxidative stress and migraine and therefore a druggable target in this disease.

## 1. Introduction

Although migraine is the second cause of disability worldwide, the accuracy of its diagnosis and the availability of therapeutic options are still unsatisfactory, despite introducing calcitonin gene-related peptide (CGRP) receptor antagonists and antibodies against CGRP and its receptor that revolutionized migraine treatment [1,2]. One of the main reasons for insufficient migraine diagnosis and treatment is the insufficiently known mechanism of its pathogenesis. This insufficiency mainly results from restricted access to the human target material and limitations of the animal models of human migraine [3]. Therefore, studies on the molecular mechanisms of migraine pathogenesis are justified. 

Sensitization of the trigeminovascular system and cortical hyperexcitability are critical mechanisms in migraine pathogenesis [4]. Neuronal sensitization due to cortical spreading depolarization/depression (CSD) is proposed as a crucial mechanism connecting migraine aura and pain onset [5]. CSD may be the mechanism of cortical hyperexcitability, and animal studies suggest that it can activate meningeal nociceptors and, in this way, contribute to headache pain and not only aura [6,7]. 

Nociceptive primary afferent nerve fiber endings express ion channel receptors that transform harmful environmental stimuli into electric signals [8]. Furthermore, such receptors may intensify the transmission of nociceptive signals to central neurons [9]. Harmful stimuli sensed by ion channel receptors on peripheral terminals may induce nociceptive signals that may cause pain and defense reactions to avoid damage [10,11]. 

Humans express 28 different transient receptor proteins (TRPs) ion channels that can be divided into several subfamilies [11]. The important role of TRPs in migraine pathogenesis is supported by the pattern of their expression as they are expressed in trigeminal neurons and brain regions affected by migraine [12]. They are signaling molecules and can change harmful impulses into pain signals. As TRPs are transmembrane proteins, they may link the extracellular environment with processes occurring within neurons. Botulinum neurotoxin type A, a drug used in chronic migraine therapy, is an example of drugs that affect TRP channels [13]. As TRP channels affect CGRP release, they may be considered to support antimigraine therapy with anti-CGRP antibodies [14]. Migraine is caused by channelopathy with an important role of the hyper-sensory brain and increased connections between sensory neurons with the involvement of several compounds, including sodium, potassium, aldosterone, cortisol, adrenaline, and glucose [15]. Therefore, TRP channels which are important in pain transmission and inflammation may play an important role in migraine pathogenesis.

Transient receptor potential vanilloid 1 (TRPV1), activated by heat and uniquely sensitive to capsaicin, is likely the most intensively studied TRP channel considered a target for pain treatment [16]. Dux et al. showed that capsaicin-sensitive TRPV1-expressing nociceptive afferent nerves contributed to the dural vasodilatory responses suggesting an important role of TRPV1 in meningeal nociception and migraine pain [17]. Transient receptor potential cation channel subfamily A member 1 (TRPA1) is another TRP channel, which was found to co-localize with TRPV1 in a subpopulation of neurons. It is therefore a candidate to contribute to migraine headaches [18]. Moreover, TRPA1 in trigeminal neurons can be activated by several factors that are considered migraine triggers [19]. Furthermore, some chemicals projected to be used or used in migraine treatment desensitize or inhibit TRPA1 [20]. Agonists of TRPA1 induced the release of CGRP, neurogenic inflammation, pain sensitivity, and behaviors typical for migraine in experimental animals [20,21]. 

While there is no convincing evidence that oxidative stress plays a substantial role in migraine pathogenesis, many migraine triggers are related to oxidative stress [22]. Experimental data in rats suggest that after CSD, oxidative stress spreads downstream within the trigeminal nociceptive system and could be involved in the coupling of CSD with the TRPA1-mediated activation of the trigeminovascular system in migraine [23]. 

Antibody deactivation of TRPA1 expressed in cortical neurons and astrocytes of rats and mice reduced cortical susceptibility to CSD in rats and decreased ipsilateral malondialdehyde levels induced by CSD [24]. Hydrogen peroxide, a canonical oxidative stress inducer, accelerated submaximal CSD induction in mouse brain slices, but treatment with an antioxidant and TRPA1 antagonist suppressed that effect. Furthermore, TRPA1 activation reversed prolonged CSD latency and reduced its magnitude via antioxidant treatment. Furthermore, the blockade of CGRP prolonged CSD latency, which was reversed by treatments with hydrogen peroxide and a TRPA agonist. It was concluded that the ROS-TRPA1-CGRP signaling played a crucial role in the cortical susceptibility to CSD. 

In this narrative review, we present information on the mutual connections between oxidative stress and migraine, the structure and functions of TRPA1, and its role in oxidative stress sensing and defense in some pathologies, including migraine. Finally, we review the arguments that TRPA1 can be a “missing element” linking oxidative stress with migraine. 

## 2. Oxidative Stress in Migraine 

Due to intense glucose metabolism, the human brain produces reactive oxygen and nitrogen species (RONS) that are byproducts of oxidative phosphorylation (OPHOS) in mitochondria. These RONS are produced in normal conditions, but their level increases with misfunctioning OPHOS due to an impaired electron transport chain (ETC). When the cellular antioxidant defense cannot cope with increased levels of RONS, the cell is in a state of oxidative stress. RONS produced in oxidative stress may damage brain macromolecules, including proteins, nucleic acids, and lipids. 

As mentioned, there is no experimental evidence of the causative role of oxidative stress in migraine; some data from animal experiments and rationales suggest that oxidative stress may be related to migraine attacks [25]. However, this relationship may not be unequivocal. 

As oxidative stress in the brain may lead to the death of brain cells and degenerative changes, the antioxidant systems may not counteract oxidative stress in the brain. Migraine was proposed as a defensive response against a threat to the brain caused by oxidative stress, and this suggestion is supported by the observation of increased antioxidant defense in migraine [25,26,27].

Energetic relationships in the brain may be critical for the role of oxidative stress in migraine pathogenesis as migraine features brain energy deficit [28]. This deficit may result from impaired energy production or increased demand during migraine attacks. Both scenarios may lead to overproduction of RONS, as impaired energy production is associated with impairment in ETC functioning, which can be linked with the overproduction of RONS by the ETC proteins. On the other hand, increased energy demand by the migraine-susceptible brain may induce increased energy production. Therefore, increased activity of the ETC proteins is also associated with increased RONS, which may contribute to oxidative stress. 

Nuclear magnetic resonance (NMR) imaging enables the direct determination of energetic relationships within the brain in the interictal and attack phases of migraine [29]. The basic indicators of energy deficit in the brain of migraine patients between and during attacks are a decreased phosphocreatine to creatine ratio and increased concentration of ADP measured with ^31^P NMR [30,31]. Accordingly, a lower ATP concentration was reported in migraine patients [32]. However, ATP does not directly provide energy but must be hydrolyzed, and that process may occur with different efficacy in different states, i.e., a particular level of ATP may result in different amounts of energy, depending on the state. Lodi et al. showed that the levels of free energy released by ATP hydrolysis were the lowest in patients with migraine stroke and the highest in patients with migraine without aura [33]. Moreover, the amount of that energy correlated with the severity of the diseases. That study also showed different, generally lower, Mg^2+^ concentrations in various headache-related states. Another study correlated the amount of energy deficit in migraine with the frequency of attacks [34].

To counteract the consequences of increased production of RONS, the cells evolved an antioxidant defense, in which the main components are antioxidant enzymes, antioxidant peptides, DNA repair, and small-weight antioxidant molecules. All these elements are reported to be affected in migraine. Decreased levels of reduced glutathione (GSH), glutathione-S-transferase (GST), and total antioxidant activity (TAC) were found in migraine patients [35]. Some variants of DNA repair genes, including X-ray repair cross complementing 1 (*XRCC1*), X-ray repair cross complementing 3 (*XRCC3*), ERCC excision repair 2, TFIIH core complex helicase subunit (*XPD, ERCC2*), ERCC excision repair 5, endonuclease (*XPG*, *ERCC5*), apurinic/apyrimidinic endodeoxyribonuclease 1 (*APE1, APEX1*), and 8-oxoguanine DNA glycosylase (*hOGG1*), involved in the repair of oxidatively damaged DNA, were associated with increased migraine risk [36]. We also presented arguments that DNA repair in migraine might be compromised because transient receptor potential melastatin 2 (TRPM2) might link DNA single-strand break repair proteins, including poly(ADP-ribose) polymerase 1 (PARP1), XRCC1, and DNA topoisomerases, with migraine-related neuroinflammation [37]. Lower levels of antioxidant vitamins and minerals in migraine patients were observed in several studies, e.g., [38,39,40]. However, two issues might be addressed considering the role of antioxidant vitamins in migraine pathogenesis. Firstly, their antioxidant activity and so their role in oxidative stress is small in comparison with antioxidant enzymes and DNA repair proteins. Secondly, many vitamins show both anti- and pro-oxidative properties, depending on the cellular environment dictated by endogenous and exogenous influences and so they may induce unexpected effects [41]. In summary, all main components of the cellular antioxidant defense can be impaired in migraine, inducing or potentiating oxidative stress. Although it is possible to mechanistically link migraine with oxidative stress and vice versa, the determination of the causal relationship between migraine and oxidative stress may depend on specific conditions and individual reactions of the brain to the stress, and oxidative stress as a causal factor may add to the factors of migraine pathogenesis, but it is not the sole reason of migraine induction or progression. 

Human studies on the role of oxidative stress in migraine pathogenesis are restricted for obvious reasons. Most of them are performed in peripheral blood, and studies in cerebrospinal fluid are rare [42]. However, many peripheral biomarkers are considered as reliable indicators of migraine (reviewed in [42]). The human studies on the brain are limited to neuroimaging and largely do not provide direct information on oxidative stress.

In summary, many studies associate migraine with oxidative stress, but the causative relationship between them has not been established so far (Figure 1). Increased energy production and increased energetic demand by migraine-sensitive brain may lead to overproduction of RONS, which may damage proteins of the antioxidant defense system. Many migraine triggers are related to oxidative stress, which can lower their threshold. 

## 3. The TRPA1 Ion Channel: Its Gene, Structure, and Migraine-Related Activities

Transient receptor potential cation channel subfamily A member 1 (TRPA1, TRP ankyrin 1, the Wasabi receptor) is a member of the 28-member TRP superfamily of Ca^2+^-permeable cation channels. The TRP channels can integrate many signaling systems, including those involving G protein-coupled receptors (GPCRs) and growth factor receptors [11]. The activity of the TRP channels is determined by their structure, which, in turn, is dictated by the expression of their genes, including alternative transcription and posttranslational modifications. TRPs can be activated by a plethora of stimuli, including mechanical, thermal, and chemical factors, such as exogenous chemical compounds, lipids, products of oxidative stress, acids and bases, pheromones, osmolarity, mechanical stimulation, light, and thermal energy [43]. 

TRPA1 is encoded by the *TRPA1* gene, located at 8q21.1, whose transcription produces 5 mRNAs translated to 3 polypeptides (https://www.ncbi.nlm.nih.gov/datasets/gene/id/8989/products/, accessed on 26 June 2024).

TRPA1 represents the TRP ankyrin (TRPA) subfamily of the TRP channel superfamily and is abundantly expressed in human cells [11]. In the nervous system, TRPA1 is mostly expressed on myelinated Aδ- and unmyelinated C-fibers of peripheral nerves. It localizes in several nerves, including the vagus and trigeminal nerve [44]. 

Cryo-electron microscopy studies revealed a three-dimensional structure of TRPA1 [45]. The channel assembles as a homotetramer and displays structural features enabling its complex regulation by many endogenous and exogenous factors (Figure 2).

Each of the human TRPA1 homotetrameric subunits contains six transmembrane domains, S1–S6, where S1–S4 create the voltage-sensing domain and S5–S6 form the central pore and selectivity filter (Figure 3). Both N- and C-terminus are embedded in the intracellular space. The elongated ankyrin repeats and the reactive lysine and cysteine residues in the N-terminal part of TRPA1 control the protein–protein interaction and its channel functions [46]. Also, the repeats are involved in the connection of TRPA1 with the cytoskeleton and its trafficking. A β-hairpin loop followed by two α-helices and the pre-S1-helix, placed immediately before the ankyrin repeats, is involved in the allosteric regulation of TRPA1 [47]. The TRP1 channel opening and closing are regulated by its agonists/antagonists in the gates formed by sidechains of amino acid residues projected into the pore (Asp 915, Ile 957, and Val 961) [48]. The C-terminus of TRPA1 contains a calcium-binding domain, which is involved in the direct activation of TRPA1 by intracellular Ca^2+^ [49].

The ubiquitous and tissue-specific expression of TRPA1 implies its many physiological functions that can be related in many aspects to its structure [50]. Therefore, TRPA1 structural or/and functional impairments may underlie its role in the pathogenesis of many diseases. The main pathologies in which TRPA1 plays an important role are central nervous system disorders, cardiovascular disorders, respiratory diseases, skin pathologies, kidney diseases, gastrointestinal tract disorders, and eye diseases. It is out of the scope of the present work to discuss all functionalities of TRPA1 and there are many recent reviews on the role of TRPA1 in physiology and pathology, e.g., [49,51]. This review will focus on the involvement of TRPA1 in migraine-related effects, first pain transmission, neurogenic inflammation, and vasodilation. 

TRPA1 plays an essential role in pain perception [52]. Nociceptors, specific nerve endings, control the generation of nociceptive pain. Noxious stimuli of various natures above the threshold of nociceptor activation stimulate the generation of action potentials that are then transmitted as pain signals. It was suggested that the phosphorylation of the p38 mitogen-activated protein kinase (MAPK1) may mediate pain perception-related activation of TRPA1. 

Several irritants, including those related to migraine triggers, may activate TRPA1 in nociceptors, evoking action potential signaling pain [21]. RONS are agonists of TRPA1, and they evoke the *N*-methyl-d-aspartate (NMDA) receptor reaction and intensify pain signaling from primary sensory neurons to projection neurons [53]. TRPA1 agonists can stimulate the release of migraine-related neuropeptides, including CGRP and substance P (SP) that may contribute to neurogenic inflammation [54]. That and other studies support the general conclusion that TRPA1 agonists and antagonists modulate the expression of TRPA1 and CGRP and the activation of CGRP by TRPA1 may be mediated by the MAPK1/2 signaling pathway [55]. Therefore, TRPA1 may be involved in pain transmission and neurogenic inflammation, which may be stimulated by its endogenous agonists produced in oxidative stress [56]. Such activation may be hampered by TRPA1 antagonists, making the molecule a druggable target in migraine and other disorders with pain and inflammation in their pathophysiology. This was confirmed in the study on a rat nitroglycerin-induced hyperalgesia model and the TRPA1 antagonist ADM_12 [57]. That study showed the critical involvement of TRPA1 in the pathogenesis of migraine and its potential to neutralize hyperalgesia at the trigeminal level.

McNamara showed the importance of TRPA1 in inflammatory nociception via the blockade of the nociceptive response evoked by formalin stimulation in experimental animals [58]. Apart from such TRPA1 acute action, a prolonged consequence of TRPA1 activity in the form of hypersensitivity induced by a harmful stimulus was observed weeks after removing that stimulus and resolving inflammation.

The TRPA1–CGRP interaction may be the key activity of TRPA1 related to migraine pain, as the involvement of CGRP released from the terminals of trigeminal neurons in migraine-related neurogenic inflammation may belong to the main mechanisms of migraine headaches [59].

Although the vascular theory of migraine is controversial and questioned in many reports, the fact that all migraine-provoking agents are vasodilators supports the significance of vascular components in migraine pathogenesis [60,61]. Furthermore, despite the lack of convincing evidence that CGRP and substance P are behind all migraine cases, they are neurogenic-dependent vasodilators [62,63]. TRPA1 was reported to localize on the endothelium of cerebral arteries, and allyl isothiocyanate (AITC), an exogenous TRPA1 activator, mediated the relaxation dependent on endothelium in rat cerebral arteries [64,65]. A local cold exposure in the hindpaw vasculature of mice resulted in transient vasoconstriction followed by vasodilatation [66]. Therefore, the activation of TRPA1 may play an important role in vasodilation, but the mechanism underlying that involvement is not completely clear. Some light was shed by a study showing that cinnamaldehyde-induced vasodilatation occurred in wild-type but not in TRPA1 knockout mice [66]. That study showed that cinnamaldehyde-mediated vasodilatation was reduced by a RONS scavenger, supporting the role of RONS in the downstream vasodilatory TRPA1-mediated response.

Accumulating evidence points to the involvement of glial cells, including astrocytes, microglia, satellite glial cells, and Schwann cells of the trigeminovascular system, trigeminal nucleus caudalis, and cortex in migraine pathogenesis [67]. The involvement of glial cells TRPA1 in pain transmission was reported in several studies, but they are not completely consistent [59]. However, De Logu et al. showed that oral intake of alcohol, a migraine trigger, might result in targeting TRPA1 in Schwann cells by acetaldehyde, an immediate product of ethanol metabolism, resulting in the generation of RONS that may activate TRPA1 in nociceptors to signal pain [68]. 

In summary, TRPA1 may display several activities in neural cells that stimulate CGRP release from neural afferents. However, these activities can also be related to neuroinflammation and vasodilation, independently of CGRP. RONS can be an important element in TRPA1 regulation in neuronal and glial cells. 

## 4. TRPA1 and Oxidative Stress

Many studies showed that TRP channels detect reactive species and induce various physiological and pathological responses, including cell death, chemokine production, and pain transduction [69]. TRP channels may sense reactive species through second messengers or via oxidative modification of cysteine residues [70]. The exact mechanism of TRPA1 activation by RONS is unknown.

Many studies on the connection between TRPA1 and oxidative stress were performed in cancer [71]. Oxidative stress and RONS play a dual role in cancer transformation and the fate of cancer cells [72]. Oxidative stress is supportively associated with cancer transformation as RONS are involved in signaling pathways important in initiating, promoting, and progressing the transformation and inducing mutations that confer cancer cells advantages over normal cells. On the other hand, oxidative stress may lead to cancer cell death. Therefore, cancer cells have evolved mechanisms to cope with oxidative stress that are different from those operating in their normal counterparts, because cancer cells must accommodate relatively high levels of RONS detrimental to normal cells. On the other hand, cancer cells must not allow stress to induce harmful effects that may lead to cell death. Therefore, the antioxidant defense system of cancer cells must be different from that typical for normal cells. To adapt to high RONS levels, cancer cells made several modifications in their structure and functions, including sulfur-based metabolism, NADPH generation, and the activity of antioxidant transcription factors [72].

TRPA1 plays an important role in regulating cancer cells’ response to oxidative stress, which contains some elements that may be applied for other than cancer pathologies. Numerous works demonstrate that TRPA1 is upregulated in several cancer types [73]. TRPA1 may send Ca^2+^ signals to activate antiapoptotic pathways or to promote mitochondrial dysfunction and apoptosis [71]. Faris et al. showed that TRPA1 was upregulated and mediated enhanced hydrogen peroxide-induced Ca^2+^ influx in metastatic colorectal cancer (mCRC) cells [74]. 4-hydroxynonenal (4-HNE), a lipid peroxidation product, was the main RONS responsible for TRPA1 activation in mCRC cells in oxidative stress. TRPA1-mediated calcium influx in response to oxidative stress results in mitochondrial overload with calcium, mitochondrial depolarization, and the activation of caspase-3/7. Therefore, TRPA1 may play multiple important roles in the connection between oxidative stress and cancer transformation, illustrating its general role as an oxidative stress sensor and modulator. In general, TRPA1 plays a similar role in the nervous system and its pathologies (reviewed in [75]).

Andersson et al. showed that TRPA1 was a sensory receptor for a broad class of chemicals related to oxidative stress in TRPA1-expressing CHO cells and sensory neurons [56]. They were hydrogen peroxide, endogenously occurring alkenyl aldehydes: 4-HNE, 4-oxo-nonenal, 4-hydroxyhexenal, and the cyclopentenone prostaglandin, 15-deoxy-Δ^12,14^-prostaglandin J_2_ (15-d-PGJ). The effect induced by hydrogen peroxide was reversed via treatment with dithiothreitol, suggesting that it mediated the formation of disulfide bonds in contrast to alkenyl aldehydes and prostaglandin and 15d-PGJ, which might form Michael adducts. Hydrogen peroxide and the naturally occurring alkenyl aldehydes and 15d-PGJ acted on a subset of isolated rat and mouse sensory neurons to induce a depolarizing inward current and increased the Ca^2+^ concentration in neurons expressing TRPA1. These effects were greatly reduced in neurons from mice with a double knockout in the *TRPA1* gene. Injection of H_2_O_2_ or 15d-PGJ2 induced a nocifensive/pain response in wild-type mice, but not in the mice with *TRPA1* knockout. Therefore, TRPA1 in sensory neurons may be activated by a wide spectrum of oxidative stress-related products with pain signaling. 

As mentioned, TRPA1 can be activated by multiple stimulations and an immediate question is what mechanism is behind such polymodality of activation by various stimuli. In particular, it is important whether this feature is displayed in the TRPA1 structure, in particular whether compounds that activate TRPA1 represent different classes of chemicals characterized by different chemical and physical properties. It was proposed that non-electrophilic complexes might interact with TRPA1 through the traditional binding pocket model in a reversible manner [76]. Reactive electrophilic compounds bind to cysteine thiols of TRPA1 N-terminus via nucleophilic attack [77]. The electrophile-induced response of TRPA1 was attenuated when codons for three cytosolic cysteine residues, C619, C639, and C663, located in the N-terminus of human TRPA1, were mutated [78]. However, these mutations did not completely inactivate TRPA1, and K708, in the close vicinity of the essential cysteines, was identified as critical for such residual activity [79]. Further mutagenesis studies showed that thiol-sensitive TRPA1 activators interacted with the specific cysteine residues C415, C422, C622, C642, C666, C174, C193, Cys 634, and C859 (mouse numbering) or lysines K620 and K710 [79,80,81]. These critical cysteines undergo disulfide-bond formation or reorganization, resulting in conformational alterations in the N-terminus and channel activation or desensitization [82,83]. Cysteine is a major target amino acid for cellular oxidation due to its high reducing property. A mouse study revealed that channel modulation results in a conformational reorganization in the N-terminal ankyrin repeats, the pre-S1 helix, the TRP-like domain, and the linker regions of the channel [84]. 

The seminal work of Takahashi et al. provided information on the pattern of TRPA1 activation measured by the intracellular calcium concentration in dependence on oxygen concentration [81]. The lowest Ca^2+^ concentration was observed in normoxia (20% O_2_), while the highest was in hypoxia (14% O_2_) and hyperoxia (86% O_2_). Therefore, this pattern can determine the reaction of TRPA1 to oxidative stress. The underlying mechanisms may result from the activity of prolyl hydroxylases and the direct oxidation of cysteine residues in the N-terminus [85]. 

Therefore, TRPA1 activation in oxidative stress is effected through the oxidation of cysteine residues. Hyperoxidation results in disulfide covalent binding between the residues and overrides proline hydroxylation, resulting in an overactive TRPA1. This leads to the generation of electrical impulses in Aδ, and C nociceptive fibers, which may activate the inflammatory cascade [85]. The involvement of TRPA1 in the inflammatory reaction in several pathological conditions, including airway inflammation, was shown by Yao et al. [55]. 

The TRPA1 molecule is not the only one that senses oxidative stress and uses cysteine thiols to neutralize the products of the stress. Consequently, the question about the relationship of TRPA1 with other elements of the antioxidant systems is justified. Many pathways of interactions of additive and synergic character can be considered, and the mutual regulation between TRPA1 and hypoxia-inducible factor 1-alpha (HIF1A) in human fibroblast-like synoviocytes is a representative example of them [86]. HIF1A in synoviocytes was activated by the nuclear factor NF-kappa-B p105 subunit (NFKB1) signaling resulting from the interaction between tumor necrosis factor (TNF) and interleukin-1 alpha (IL1A). Activated HIF1A bound one of the hypoxia response element-like motifs functioning as enhancers in the *TRPA1* gene promoter. Therefore, HIF1A may regulate inflammation through the interaction with TRPA1, illustrating an interplay between TRPA1 and another oxidative stress sensor.

In summary, TRPA1 is a sensor and modulator of oxidative stress, and because such stress is implicated in many pathologies, it plays an important role in the pathogenesis of several disorders. The mechanism behind the involvement of TRPA1 in sensing and modulating oxidative stress is primarily based on the modifications of cysteines and lysines in the N-terminus of TRPA1 and modulating the activity of oxidative stress-related enzymes. Consequently, TRPA1 may be considered a diagnostic and therapeutic target in oxidative stress-related diseases.

## 5. Role of TRPA1 in Oxidative Stress in Migraine 

TRPA1 was identified as a sensor of RONS at the sites of inflammation or tissue damage in several disorders, including inflammatory and neuropathic pain and migraine [21]. Therefore, it is important to recognize molecular pathways leading from oxidative stress to migraine mediated by TRPA1. In particular, TRPA1-mediated consequences of oxidative stress-related migraine triggers may significantly contribute to migraine pathogenesis.

Cigarette smoke enhances oxidative stress as it increases RONS production and compromises the antioxidant system [87]. Cigarette smoke is considered a migraine trigger, affects the occurrence of headaches in migraine patients, and is reported to increase the frequency of cluster headaches [88]. The prevalence of smoking is higher in migraine patients and they reported more severe migraine attacks than their non-smoking counterparts. Also, smoking is reported to increase the chance of migraine-related consequences, including stroke [89]. Although cessation of smoking has a beneficial effect on the health of migraine patients, they have a more difficult time quitting smoking than persons without migraine. The influence of smoking and quitting on the health of migraine patients is not widely represented in research, and the literature and mechanistic explanation of the connection between smoking and migraine is not exactly known. 

Oxidative stress evokes peroxidation of lipids of the plasma membrane, producing acrolein, an α,β-unsaturated aldehyde, and an electrophile present in cigarette smoke [90]. TRPA1 was reported to play a major role in the morbidity and mortality caused by inhalation of acrolein [91,92]. The activation of TRPA1 was reported to mediate the neurogenic and inflammatory action of acrolein on sensory nerve endings [90,93]. For this reason, chemicals in cigarette smoke induce airway inflammatory responses through stimulation of TRPA1 channels expressed in vagal sensory nerve endings [94]. Acrolein induced CGRP release and increased the meningeal blood flow in dependence on TRPA1 activation [95]. Therefore, it is justified to hypothesize that acrolein, due to its capability to induce TRPA1 activation and CGRP release, mediates neurogenic inflammation and headache evoked by cigarette smoke [21]. This mechanism can be generalized to other chemicals, including 2-chlorobenzalmalononitrile, a tear gas inducing headache [96].

Oxidative stress in neurons is associated with the overproduction of RONS, which, at least in part, may diffuse through the mitochondrial and then neuronal membrane and may become external triggers to activate Ca^2+^-permeable cation channels encoded by the TRP gene superfamily [69]. The TRP proteins may transduce the RONS signal into a neural signal of oxidative stress as argued for hydrogen peroxide [97]. TRP ion channels can also be activated by lipid peroxides that are products of the interaction of RONS with cellular membranes and can diffuse out of the cell. Lipid peroxides are characterized by longer lifetimes than RONS and therefore may contribute to persistent activation of TRPA1. TRPA1 localizes in the perivascular meninges on nociceptive nerve endings and may sensitize meningeal nociceptors, and second-order trigeminal neurons, which may be important in migraine [98]. Therefore, TRPA1 can be considered a sensor of oxidative stress that may induce pain signaling on activation.

Nitrogen oxide has a central position in migraine pathogenesis, as nitroglycerine (NTG), a NO donor, is a prototype of migraine-inducing agents [99]. However, NO, which contributes to oxidative stress, may be related not only to the pathogenesis of migraine, but tension-type headache, and cluster headache as well (reviewed in [100]). NO can induce vasodilation of cranial arteries, which can be attributed to the ability of NTG to evoke headaches [101]. Bellamy et al. showed that NO triggered signaling mechanisms within the trigeminal ganglia neurons that stimulate CGRP synthesis and release [102]. However, it is not known whether that effect could be directly related to TRPA1, although TRPA1 activation was a mechanism behind the stimulation of sensory neurons by NO. Subsequently, Eberhardt et al. demonstrated that H_2_S and NO cooperatively regulated vascular tone by activating a neuroendocrine HNO-TRPA1-CGRP signaling pathway [103]. Nitroxyl, HNO, is a one-electron redox form of NO. Therefore, TRPA1 may be a central element of the signaling pathway from oxidative stress and NO production to CGRP release, which may play a critical role in headache induction. However, the exact mechanism underlying the role of TRPA1 in this effect is not known. Glyceryl nitrate (GTN) is a potent vasodilator and another NO donor [104]. Marone et al. showed that TRPA1 and NDPH oxidase 1/2 (NOX1/2) supported an autocrine pathway within the neuronal cell bodies of trigeminal ganglia to sensitize meningeal nociceptors and second-order trigeminal neurons to provoke periorbital allodynia and could be important for migraine-like headaches induced by GTN [98]. That study suggested that the activation of TRPA1 might be induced by an NO-mediated interaction of GTN with TRPA1 and RONS production due to an increased activity of NADPH oxidase in the trigeminal ganglion.

Cortical spreading depression evokes an early severe decrease in the synthesis of NADH, weakening the antioxidant system [105]. However, NADH binding and local hypoxia cause overproduction of RONS by ETC. Furthermore, CSD may directly induce oxidative stress, as shown by Shatillo et al. in the rat trigeminal nociceptive system, in which TRPA1 mediated the pro-oxidative action of hydrogen peroxide [23]. 

In summary, TRPA1 may connect oxidative stress with pain induction and transmission in migraine. TRPA1 senses oxidative stress products, sends the signal to sensory neurons, induces CGRP release, and activates the trigeminovascular system to evoke headaches (Figure 4). When oxidative stress is induced within neurons in excessive energy production or ETC impairment in mitochondria, RONS may diffuse to the mitochondrial and plasmatic membrane and induce similar effects as external oxidative stress. Nitric oxide is an important mediator of the TRPA1-CGRP pathway.

## 6. Conclusions and Perspectives

The role of CGRP in migraine pathogenesis is well established, and therefore, the involvement of TRPA1 in the CGRP release from nerve endings allows us to consider TRPA1 as another key molecule in migraine, the more that TRPA1 is sensitized by numerous migraine triggers and inhibited by analgesics [59]. Also, the antimigraine action of some traditional medicines, including certain herbs, can be explained by their interaction with TRPA1 [106]. As TRPA1 can sense RONS and send pain signals to the brain regions important for migraine, it may be a missing link between oxidative stress and migraine. Consequently, TRPA1 is considered a druggable target in migraine, other primary headaches, and other pain-related disorders [16]. However, the role of TRPA1 activation is not limited to oxidative stress and migraine, as it can be activated by many factors not related to oxidative stress and disorders other than migraine. Cinicaltrails.gov lists 19 clinical trials with TRPA1, the majority of which are related to pain (https://clinicaltrials.gov/search?term=trpa1, accessed on 27 July 2024). However, none are directly related to migraine, although one deals with neurogenic inflammation and pain. 

The introduction of CGRP receptor antagonists and antibodies against CGRP and its receptor into pharmaceuticals marked revolutionized migraine treatment [2]. However, although these drugs are in general well-tolerated, not all migraine patients treated with these drugs show a satisfactory response—clinical studies indicate that about one-third of patients showed less than 50% resolution of symptoms [107,108,109]. Therefore, new therapeutic targets are needed and TRPA1 is a prime candidate to take that role. However, due to having several functions other than pain transmission, the viability of TRPA1 as a drug target in migraine requires further studies.

As mentioned, TRPA1 may be a link between oxidative stress and migraine. It is difficult to find a disease that would be not related to oxidative stress, but in many cases, that relationship is supported by associations and speculations only and not by any specific mechanism. TRPA1, as an oxidative stress sensor, CGRP inducer, and pain transmitter, is representative of the specific mechanism behind the association of oxidative stress and migraine.

Another mechanism of TRPA1 activation is its modulation by G protein-coupled receptors (GPCRs) through the second-messenger signaling cascades [110]. TRPA1, like other TRP channels, is a downstream effector of GPCR nociceptive and pruritogenic signaling and consequently forms the GPCR-TRPA1 axis to sense pain, neurogenic inflammation, and analgesia [111]. It is important to relate this mode of TRPA1 activation to oxidative stress, especially as emerging data now suggest that GPCRs could function as the sensors of stresses, including oxidative stress [112].

Migraine is not categorized as a neurodegenerative disease, but emerging evidence points to the association of migraine with morphological changes in the brain [113]. Such changes may be underlined by damage to neurons and other brain cells as well as cells of the brain stem and cerebellum [114,115]. Reactive oxygen and nitrogen species produced during migraine-associated oxidative stress may damage cellular macromolecules, including DNA/RNA, proteins, and lipids, resulting in damage that can be observed with neuroimaging techniques [37].

Females show 2–3 times higher migraine prevalence and more severe headache attacks than males and the reason for that sexual dimorphism is not completely known [116]. Transient receptor channels are among other candidates behind that relationship, especially since they may be regulated by sex hormone steroids, as shown for TRPA3 [117]. Therefore, the role of TRPA1 in sexual dimorphism in migraine prevalence might be investigated in further research as justified by other studies on the role of TRP channels in that dimorphism [118,119]. However, the gender dimension is not considerably addressed in migraine studies. Lemos et al. showed that gender was a risk factor in migraine, with females at a higher risk [120]. The gender dimension in migraine requires further studies, especially due to the inadequacy of animal models in this regard. Whether TRPA channels should be included in such studies is an open question.

Although the development of neuroimaging has provided access to the migraine target tissue, it has not supplied precise information on anatomical and neurochemical pathways responsible for migraine attacks. Also, genetic studies did not identify the genome variants that might determine migraine susceptibility. Biomolecular studies on migraine pathogenesis may shed light on the mechanism of information transfer between migraine genotype and phenotype, influenced by endogenous and exogenous environmental factors. Such a palette of research should also contain pharmacological as well as observational and interventional clinical studies to verify the suitability of TRPA1 as a druggable target in migraine [121].

## Figures and Tables

**Figure 1 molecules-29-03385-f001:**
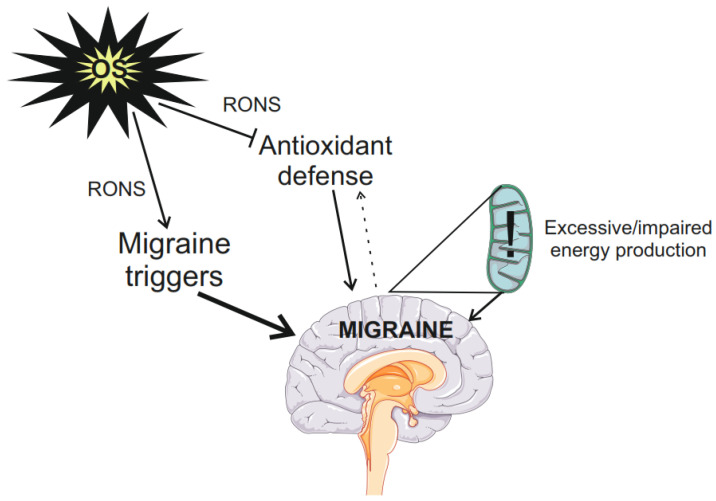
Oxidative stress (OS) in migraine. Oxidative stress may be attributed to external or internal factors, but both result in the overproduction of reactive oxygen and nitrogen species (RONS) that may damage biomolecules. RONS may modulate migraine triggers and lower their thresholds (bold arrow), RONS may damage proteins of the cellular antioxidant defense lowering its efficacy, but on the other hand, migraine is reported to stimulate that system (dotted arrow). Increased energy production or/and demand by migraine susceptible brain leads to overproduction of RONS that may damage the protein of the electron transport chain (ETC) in mitochondria (exclamation mark). Impaired ETC produces more and more RONS (“mitochondrial vicious cycle”). Parts of this figure were drawn by using pictures from Servier Medical Art. Servier Medical Art by Servier is licensed under a Creative Commons Attribution 3.0 Unported License (https://creativecommons.org/licenses/by/3.0/ accessed on 26 June 2024).

**Figure 2 molecules-29-03385-f002:**
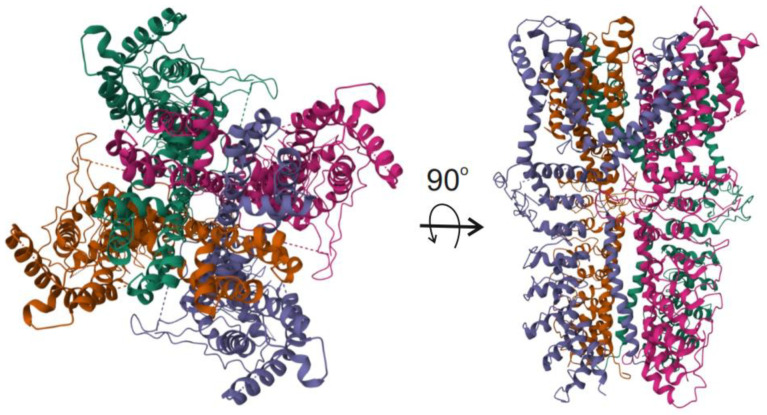
Ribbon representation of transient receptor potential cation channel subfamily A member 1 (TRPA1), which usually functions as a homotetramer. Each monomer is represented with a different color. Taken from the RCSB Protein Data Bank: 3J9P: C.E. Paulsen, J.P. Armache, Y. Gao, Y. Cheng, D. Julius. Structure of the TRPA1 ion channel determined via electron cryo-microscopy. Nature (2015) https://doi.org/10.2210/pdb3J9P/pdb (accessed on 26 June 2024) under the CC0 1.0 Universal (CC0 1.0) Public Domain Dedication.

**Figure 3 molecules-29-03385-f003:**
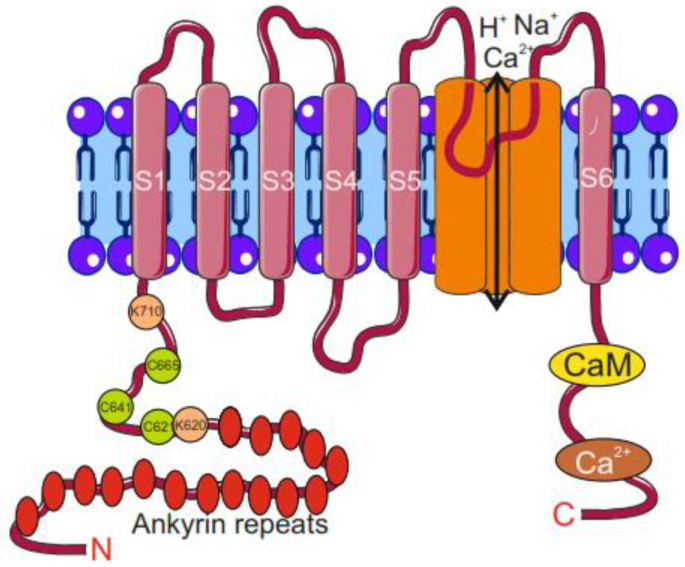
Structure of transient receptor potential cation channel subfamily A member 1 (TRPA1) channel monomer. TRPA1 monomer contains 6 transmembrane domains, S1–S6, and its N- and C-terminus are embedded in the cytosol. The S5–S6 region forms the central pore and selectivity filter for ions to enter/exit the cell. The N-terminus contains 16 ankyrin repeats and reactive cysteine and lysine residues. Calmodulin (CaM) and Ca^2+^-binding sites are located at the C-terminus. Parts of this figure were drawn by using pictures from Servier Medical Art. Servier Medical Art by Servier is licensed under a Creative Commons Attribution 3.0 Unported License (https://creativecommons.org/licenses/by/3.0/ accessed on 26 June 2024).

**Figure 4 molecules-29-03385-f004:**
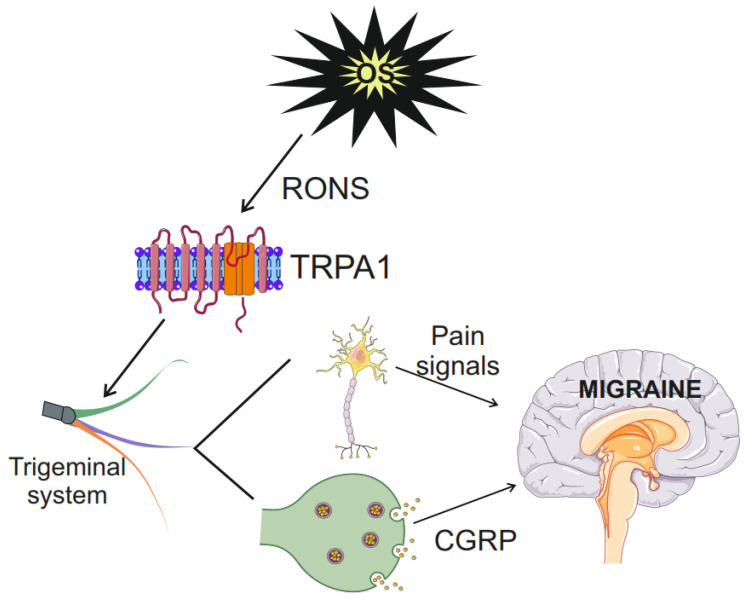
Transient receptor potential cation channel subfamily A member 1 (TRPA1) channel links oxidative stress (OS) with migraine. Oxidative stress resulting from the action of exogenous factors, including migraine triggers or excessive energy production in the migraine susceptible brain, is the source of reactive oxygen and nitrogen species (RONS). TRPA1, represented by its monomer, senses RONS and sends pain signals to sensory neurons, including the trigeminal nerve neurons, essential in migraine pathogenesis. TRPA1 induces the release of calcitonin gene-related peptide (CGRP), a key molecule in migraine. Parts of this figure were drawn by using pictures from Servier Medical Art. Servier Medical Art by Servier is licensed under a Creative Commons Attribution 3.0 Unported License (https://creativecommons.org/licenses/by/3.0/ accessed on 26 June 2024).

## Data Availability

Not applicable.

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
