# Peer review of "The TRPA1 Ion Channel Mediates Oxidative Stress-Related Migraine Pathogenesis"

_molecules, 2024, doi:10.3390/molecules29143385_

Round 1
Reviewer 1 Report
Comments and Suggestions for Authors
The current review discusses the correlation between TRPA1 ion channel and oxidative stress in migraine pathogenesis. The review is informative and well organized but the specific pathway that links oxidative stress with TRPA1 that causes migraine is not clearly stated. References were recent, and the conclusion summarizes all points discussed in the review. However, some points need to be clarified, as follows.
1. In the abstract “Migraine has been proposed as a superior mechanism to of the brain to face” Please revise the sentence.
2. The CGRP abbreviation must be written as full name in the abstract.
3. In line number 206 “(https://www.ncbi.nlm.nih.gov/da- 206tasets/gene/id/8989/products/, accessed June 26, 2024)” Is that a reference or what?
4. The figures must be mentioned in the text before their location, not after it.
5. In line number 390, “The involvement of TRPA1 in the inflammatory reaction in several pathological conditions, including airway inflammation, was shown [55]”. When the authors talk about another study’s results by “was shown”, they must mention the authors’ name and the year not just the reference number. Please revise.
6. The abbreviation with its full name must be written once in the manuscript then the abbreviation used alone.
7. In line number 497, “Figure 1. Excessive energy production in the brain may induce oxidative stress associated with overproduction of reactive oxygen species (ROS) in brain neurons” Is that a title of the figure or explanation?
8. In lines 499-500, “ROS, here exemplified by H2O may diffuse out of neurons ………..”, Does the author mean H2O or H2O2?
Author Response
Comment: The current review discusses the correlation between TRPA1 ion channel and oxidative stress in migraine pathogenesis. The review is informative and well organized but the specific pathway that links oxidative stress with TRPA1 that causes migraine is not clearly stated. References were recent, and the conclusion summarizes all points discussed in the review. However, some points need to be clarified, as follows.
Answer: Thank you.
Comment 1. In the abstract “Migraine has been proposed as a superior mechanism to of the brain to face” Please revise the sentence.
Answer: We have changed that sentence into:
“Migraine has been proposed as a superior mechanism of the brain to face…”
Comment 2. The CGRP abbreviation must be written as full name in the abstract.
Answer: The CGRP's full name is written in the first sentence of the abstract.
Comment 3. In line number 206 “(https://www.ncbi.nlm.nih.gov/da- 206tasets/gene/id/8989/products/, accessed June 26, 2024)” Is that a reference or what?
Answer: It is an Internet website cited according to the Molecules Instruction for Authors
Comment: 4. The figures must be mentioned in the text before their location, not after it.
Answer: According to the Molecules Instruction for Authors: “All Figures, Schemes and Tables should be inserted into the main text close to their first citation” and there is not an imperative to mention a figure before its placement.
Comment: 5. In line number 390, “The involvement of TRPA1 in the inflammatory reaction in several pathological conditions, including airway inflammation, was shown [55]”. When the authors talk about another study’s results by “was shown”, they must mention the authors’ name and the year not just the reference number. Please revise.
Answer: We have followed that remark, but we cannot cite a year of publication as such notation does not adhere to the Molecules Instruction for Authors (the reference list is not in alphabetical order). E.g., we have changed the phrase:
“It was shown that … [100].”
into:
“Smith et al. showed that … [100].”
Comment 6. The abbreviation with its full name must be written once in the manuscript then the abbreviation used alone.
Answer: We have again used full names for items previously defined when they opened a sentence.
Comment 7. In line number 497, “Figure 1. Excessive energy production in the brain may induce oxidative stress associated with overproduction of reactive oxygen species (ROS) in brain neurons” Is that a title of the figure or explanation?
Answer: This fragment is redundant, and we have removed it from the revised manuscript (it remained from a draft version).
Comment 8. In lines 499-500, “ROS, here exemplified by H2O may diffuse out of neurons ………..”, Does the author mean H2O or H2O2?
Answer: This is in the fragment we have removed in the response to the previous Comment .
Reviewer 2 Report
Comments and Suggestions for Authors
The manuscript, entitled The TRPA1 Ion Channel Mediates Oxidative Stress-Related Migraine Pathogenesis is an important Review in its content, even if the topic is not recent.
It is well done and the figures are also suggestive. However, I would like to add some small comments:
Where appropriate, authors are encouraged to include a section addressing sex and gender considerations in their manuscripts.
1) Artero-Morales, Maite, Sara González-Rodríguez, and Antonio Ferrer-Montiel. "TRP channels as potential targets for sex-related differences in migraine pain." Frontiers in molecular biosciences 5 (2018): 380–116.
2) Cabañero, David, et al., "ThermoTRP channels in pain sexual dimorphism: new insights for drug intervention." Pharmacology & Therapeutics 240 (2022): 108297.
Where appropriate or in lines 116-117, authors are encouraged to include, the relationship between oxidative stress and neurodegenerative diseases , as suggested by these two manuscripts.
1) Meleleo D, Picciarelli V. Effect of calcium ions on human calcitonin. Possible implications for bone resorption by osteoclasts. Biometals. 2016 Feb;29(1):61-79. doi: 10.1007/s10534-015-9896-y. Epub 2015 Nov 23. PMID: 26596282.
2) Övey, Ä°. S., & NazıroÄŸlu, M. (2020). Effects of homocysteine ​​and memantine on oxidative stress related TRP cation channels in in-vitro model of Alzheimer's disease. Journal of Receptors and Signal Transduction, 41(3), 273–283. https://doi.org/10.1080/10799893.2020.1806321
I found this review, and I think the authors can improve it like this.
Jiménez-Jiménez, F.J., Alonso-Navarro, H., García-Martín, E. et al. Oxidative Stress and Migraine. Neurobiol (2024). https://doi.org/10.1007/s12035-024-04114-7
Please ensure to carefully review the sections of the manuscript that have been highlighted in the iThenticate report.
Author Response
Comment: The manuscript, entitled The TRPA1 Ion Channel Mediates Oxidative Stress-Related Migraine Pathogenesis is an important Review in its content, even if the topic is not recent.
It is well done and the figures are also suggestive. However, I would like to add some small comments:
Answer: Thank you.
Comment: Where appropriate, authors are encouraged to include a section addressing sex and gender considerations in their manuscripts.
1) Artero-Morales, Maite, Sara González-Rodríguez, and Antonio Ferrer-Montiel. "TRP channels as potential targets for sex-related differences in migraine pain." Frontiers in molecular biosciences 5 (2018): 380–116.
2) Cabañero, David, et al., "ThermoTRP channels in pain sexual dimorphism: new insights for drug intervention." Pharmacology & Therapeutics 240 (2022): 108297.
Answer: We have added the following fragment to the concluding section:
“Females show 2-3 times higher migraine prevalence and more severe headache attacks than males and the reason for that sexual dimorphism is not completely known [116]. Transient receptor channels are among other candidates behind that relationship, especially since they may be regulated by sex hormone steroids as shown for TRPA3 [117]. Therefore, the role of TRPA1 in sexual dimorphism in migraine prevalence might be investigated in further research as justified by other studies on the role of TRP channels in that dimorphism [118,119]. However, the gender dimension is not considerably addressed in migraine studies. Lemos et al. showed that gender was a risk factor in migraine with females at a higher risk [120]. The gender dimension in migraine requires further studies, especially due to the inadequacy of animal models in this regard. Whether TRPA channels should be included in such studies is an open question.”
with new references:
- Ge, R.; Chang, J.; Cao, Y. Headache disorders and relevant sex and socioeconomic patterns in adolescents and young adults across 204 countries and territories: an updated global analysis. The Journal of Headache and Pain 2023, 24, 110, doi:10.1186/s10194-023-01648-4.
- Persoons, E.; Kerselaers, S.; Voets, T.; Vriens, J.; Held, K. Partial Agonistic Actions of Sex Hormone Steroids on TRPM3 Function. International journal of molecular sciences 2021, 22, doi:10.3390/ijms222413652.
- Artero-Morales, M.; González-Rodríguez, S.; Ferrer-Montiel, A. TRP Channels as Potential Targets for Sex-Related Differences in Migraine Pain. Front Mol Biosci 2018, 5, 73, doi:10.3389/fmolb.2018.00073.
- Cabañero, D.; Villalba-Riquelme, E.; Fernández-Ballester, G.; Fernández-Carvajal, A.; Ferrer-Montiel, A. ThermoTRP channels in pain sexual dimorphism: new insights for drug intervention. Pharmacol Ther 2022, 240, 108297, doi:10.1016/j.pharmthera.2022.108297.
- Lemos, C.; Alonso, I.; Barros, J.; Sequeiros, J.; Pereira-Monteiro, J.; Mendonça, D.; Sousa, A. Assessing risk factors for migraine: differences in gender transmission. PloS one 2012, 7, e50626, doi:10.1371/journal.pone.0050626.
Comment: Where appropriate or in lines 116-117, authors are encouraged to include, the relationship between oxidative stress and neurodegenerative diseases , as suggested by these two manuscripts.
1) Meleleo D, Picciarelli V. Effect of calcium ions on human calcitonin. Possible implications for bone resorption by osteoclasts. Biometals. 2016 Feb;29(1):61-79. doi: 10.1007/s10534-015-9896-y. Epub 2015 Nov 23. PMID: 26596282.
2) Övey, Ä°. S., & NazıroÄŸlu, M. (2020). Effects of homocysteine ​​and memantine on oxidative stress related TRP cation channels in in-vitro model of Alzheimer's disease. Journal of Receptors and Signal Transduction, 41(3), 273–283. https://doi.org/10.1080/10799893.2020.1806321
Answer: The role of oxidative stress in neurodegenerative diseases deserves a separate review. Moreover, migraine is not considered a typical neurodegenerative disease. Therefore, we have added the following fragment to the concluding section:
“Migraine is not categorized as a neurodegenerative disease, but emerging evidence points to the association of migraine with morphological changes in the brain [113]. Such changes may be underlined by damage to neurons and other brain cells as well as cells of the brain stem and cerebellum [114,115]. Reactive oxygen and nitrogen species produced during migraine-associated oxidative stress may damage cellular macromolecules, including DNA/RNA, proteins, and lipids, resulting in damage that can be observed with neuroimaging techniques [37].”
with new references:
- Ashina, S.; Bentivegna, E.; Martelletti, P.; Eikermann-Haerter, K. Structural and Functional Brain Changes in Migraine. Pain Ther 2021, 10, 211-223, doi:10.1007/s40122-021-00240-5.
- Kruit, M.C.; Launer, L.J.; Ferrari, M.D.; van Buchem, M.A. Brain stem and cerebellar hyperintense lesions in migraine. Stroke 2006, 37, 1109-1112, doi:10.1161/01.STR.0000206446.26702.e9.
- Kruit, M.C.; van Buchem, M.A.; Launer, L.J.; Terwindt, G.M.; Ferrari, M.D. Migraine is associated with an increased risk of deep white matter lesions, subclinical posterior circulation infarcts and brain iron accumulation: the population-based MRI CAMERA study. Cephalalgia : an international journal of headache 2010, 30, 129-136, doi:10.1111/j.1468-2982.2009.01904.x.
Comment: I found this review, and I think the authors can improve it like this.
Jiménez-Jiménez, F.J., Alonso-Navarro, H., García-Martín, E. et al. Oxidative Stress and Migraine. Neurobiol (2024). https://doi.org/10.1007/s12035-024-04114-7
Answer: Thank you.
Comment: Please ensure to carefully review the sections of the manuscript that have been highlighted in the iThenticate report.
Answer: We rely on an antiplagiarism check made by the editorial team.